# Independent and joint associations of volume and intensity of physical activity on cognitive impairment among middle-aged and elderly Chinese adults: A national longitudinal study

Wei Yin[1], Yan Han[2], Hongmei Sun [3]¤*

1 China National Basketball Academy, Shandong Sport University, Jinan, Shandong, China, 2 School of Sport Communication and Information Technology, Shandong Sport University, Jinan, Shandong, China, 3 College of Sport and Health, Shandong Sport University, Jinan, Shandong, China

¤ Current Address: College of Sport and Health, Shandong Sport University, Jinan 250102, Shandong, China

* 511633810@qq.com

## Abstracts

### Background

The objective of this study was to explore the longitudinal relationship between the volume and intensity of physical activity (PA) and their combined risk for cognitive impairment (CI).

### Methods

The study included 10,174 participants from the 2011-2018 CHARLS cohort. Pennsylvania and CI were assessed using self-reported questionnaires. Statistical analyses were performed using the Cox regression model.

### Results

After adjusting for all covariates, the risk of CI was 14% lower in subjects with physical activity volume (PAV) ≥600 Metabolic Equivalent of Task (MET)-min/week compared to those with insufficient PA (HR: 0.86). The risk was reduced by 38% for subjects with PAV of 1800−2999 MET-min/week (HR: 0.62). Regarding the intensity of PA, the risk of CI was reduced by 25% for a proportion of 0.25–0.5 of (moderate to vigorous PA) MVPA to PAV (HR: 0.75) compared to a proportion of 0–0.25. Regardless of PAV, the risk of CI was lowest when the proportion of moderate to vigorous PA(MVPA) to PAV was 0.25–0.5, and 0.5–0.75 for the proportion of light-intensity physical activity (LPA) to PAV.

**Data availability statement:** Data available in a publicly accessible repository. https://charls.charlsdata.com/pages/data/111/zh-cn.html.

**Funding:** The author(s) received no specific funding for this work.

**Competing interests:** The authors declare no competing interests.

## Conclusion

The PAV 1800−2999 MET-min/week and maintaining a proportion of LPA to PAV of 0.5–0.75, or a proportion of MVPA to PAV of 0.25–0.5, are more effective in reducing the risk of CI. Policy implications should prioritize tailored physical activity strategies for individuals over 65, emphasizing low-intensity activities, safe high-intensity training, and the development of accessible urban facilities, in line with WHO guidelines.

## Introduction

Cognitive impairment (CI) refers to a significant decline or dysfunction in memory, learning, attention, language, executive function, and other cognitive domains [1], which has become a major public health problem worldwide [2–4]. According to current estimates, approximately 57 million individuals globally are living with dementia, with projections indicating this figure may rise to 152 million by 2050 [5]. Notably, China is home to about 25% of the worldwide dementia patient population [6]. The Lancet Committee highlights that dementia onset may occur as early as middle age [7], a concern particularly relevant to China where cognitive impairment prevalence is notably high among adults aged 45 years and older [8]. Given that cognitive impairment serves as a transitional state between normal aging and dementia [9], developing scientifically validated methods for early identification and risk reduction has become a crucial public health priority both in China and globally.

With no effective cure, management and intervention of underlying risk factors are essential to prevent or delay cognitive decline [10]. Research indicates that lifestyle factors, particularly physical activity (PA), are associated with cognitive function in older adults [11,12]. The World Health Organization (WHO) recommends 150–300 minutes of moderate-intensity physical activity per week, 75–150 minutes of vigorous-intensity physical activity, or some equivalent combination of moderate-intensity and vigorous-intensity aerobic physical activity per week to gain health benefits such as improvements of cognitive health [13]. Systematic reviews and meta-analyses demonstrate that PA can reduce or delay risk factors such as obesity, diabetes, and hypertension, which are known to influence cognitive decline [14,15]. Additionally, moderate to vigorous PA(MVPA) is associated with the maintenance or improvement of cognitive function and brain integrity [11] and a reduction in the risk of cognitive decline [16]. Furthermore, it can enhance or even reverse cognitive performance [4,17] and alleviate dementia symptoms in patients with mild cognitive impairment [18,19].

To the best of our knowledge, specific guidelines for PA to prevent the onset of cognitive impairment remain absent. Furthermore, the causal relationship between PA and cognitive functioning is still a subject of debate in the existing research. One study observed engaging in high-intensity PA may lead to cognitive decline in middle-aged and older adults [20], whereas another study found that moderate-to-high intensity aerobic physical activity did not slow cognitive impairment in people with mild-to-moderate dementia [21]. Light-intensity physical activity has been

demonstrated to confer protective benefits against cognitive decline in older adults [22]. Furthermore, a cross-sectional investigation revealed that maintaining an exercise volume of 1,800–2,999 MET-min/week was significantly associated with optimal cognitive performance among elderly populations [23]. However, the optimal volume and intensity of PA necessary for preventing and controlling cognitive impairment remain undetermined [24], and the combined effects of PA intensity and volume have yet to be investigated. Evidence suggests that there are age and sex differences in the effects of physical activity on brain volume [25,26]. Based on this, this study utilized a nationally representative sample from the China Health and Retirement Longitudinal Study (CHARLS) to explore the longitudinal relationship between volume and intensity of PA and their combined risk for CI.

## Methods materials and methods

### Participants and study design

The data for this study are derived from CHARLS, a prominent initiative managed by the National Development Research Institute (NDRI) of Peking University. CHARLS aims to gather high-quality microdata representing households and individuals aged 45 and above in China. This dataset facilitates an in-depth analysis of population aging in China. The national baseline survey of CHARLS was conducted in 2011 and encompassed 150 district units, 450 village units, and 17,000 individuals across approximately 10,000 households. These samples are tracked every two or three years. The CHARLS questionnaire includes individual basic information, health status and functioning, physical measurements, and community-level information. The health status and functioning section encompasses PA, chronic diseases, and cognitive functioning, providing relevant research variables for this study. The CHARLS survey project was approved by the Biomedical Ethics Committee of Peking University (IRB00001052–11015), and all participants were required to sign informed consent.

Survey data from 2011–2018 were combined by identity document, resulting in a longitudinal survey of 13,263 data. There were 489 data with positive cognitive impairment at baseline. Twenty five with missing data in PA and cognitive impairment data at baseline were excluded, and 1487 with missing data in longitudinal cognitive impairment data were excluded, leaving 11262 data. 1088 data randomly missing in any variable such as age, gender, height, weight, alcohol consumption, smoking and chronic disease medication were deleted, and 10174 data were finally included in the final statistical analysis. (see Fig 1)

### Physical activity

Physical activity was measured by the International Physical Activity Questionnaire (IPAQ). In each of the four surveys from 2011 to 2018, participants were specifically questioned regarding the duration, frequency (measured in days), and intensity (whether vigorous, moderate, or mild) of physical activities they regularly undertook for at least 10 minutes during a typical week. Based on other scholars' method [27], we transformed the time ranges by taking the middle value. The results of weekly exercise for different PAs are expressed by the product of PA frequency (how many days per week) and duration (how many hours per day), i.e., PAV = PA frequency × PA duration, and finally, the PAV (physical activity volume) score can be expressed in metabolic equivalents (METs) [28]. The MET score was calculated as follows: PAV = 8.0 × duration of vigorous-intensity physical activity (VPA) per week + 4.0 × duration of Moderate-intensity physical activity (MPA) per week + 3.3 × duration of light-intensity physical activity (LPA) per week [28]. According to IPAQ, Insufficient PA is defined as the PAV less than 600 MET minutes/week [29], therefore, we elected to divide the results according into two main categories (0–599 and ≥600) and seven subcategories in terms of MET minutes/week (0–599, 600–1199, 1200–1799, 1800–2999, 3000–5999, 6000–8999 and ≥9000). In addition, we divided the PAV MET by the MPA-MET and the LPA-MET, respectively. The proportion of MVPA/LTP to PAV categories are 0–0.25, 0.25–0.5, 0.5–0.75 and 0.75–1.

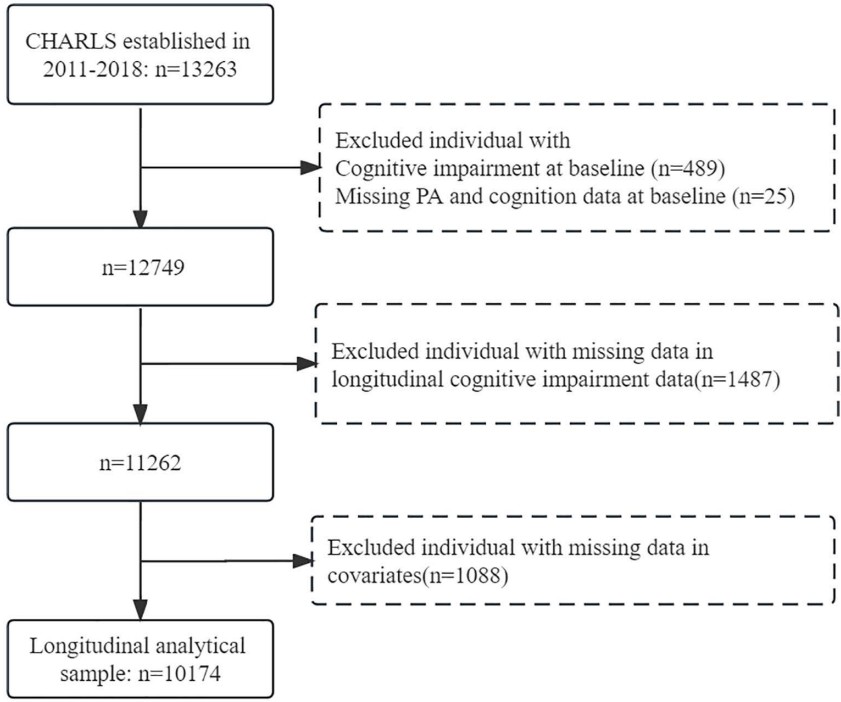

**Fig 1. The screening flowchart for the study population.**

## Cognitive impairment

Cognitive impairment was assessed based on two cognitive functions: episodic memory and executive function. Episodic memory was measured by immediate and delayed recall of words. Memory ability was evaluated by presenting 10 words to each subject, recording the number of correctly recalled words immediately (immediate word recall score) and after 4 minutes (delayed word recall score). This was quantified using the average of immediate and delayed word recall scores, which ranged from 0 to 10 [30]. Executive function was assessed using components and graphical plotting of mental status questions derived from the Telephone Interview for Cognitive Status (TICS) [31]. The TICS is a reliable and valid screening tool for cognitive impairment in older adults similar to the Mini-Mental State Examination (MMSE) [32]. The executive function score was calculated based on the total score of TICS and figure drawing, ranging from 0 to 11. The cognitive functioning score, calculated as the sum of episodic memory and executive functioning scores, ranges from 0 to 21, with higher scores indicating better cognitive functioning. Based on existing studies [33], a total score below 6 was defined as cognitive impairment.

## Covariates

According to previous articles [27,34], covariates are primarily categorized into the following groups: 1) Personal variables: age, gender (male, female), education level (high school or below, undergraduate or higher), and residence (rural, urban, or other); 2) Health status-related variables: body mass index (BMI) are categorized as follows: underweight (<18.5), normal weight ($18.5 \leq BMI < 25$), overweight ($25 \leq BMI < 30$), and obese ($BMI \geq 30$); 3) Lifestyle variables, smoking status (current smoker, former smoker, never smoker) and alcohol consumption (current drinker, former drinker, never drinker); 4) Chronic disease conditions: hypertension (no, yes), hyperlipidemia (no, yes), diabetes (no, yes), and lung disease (no, yes).

## Statistical analyses

Participants' baseline characteristics were compared between the PAV level (Insufficient PA vs Sufficient PA), categorical variables were expressed as percentages, and categorical characteristics were compared between groups using the χ2 test. Hazard ratios (HRs) and 95% confidence intervals were calculated employing the Cox proportional hazards model to examine the association between various PA volume and intensity subgroups and the risk of cognitive impairment. To evaluate the potential confounding effects of different covariates on the relationship between PA and cognitive impairment risk, we developed three distinct models. Model I was adjusted for age and gender; Model II was adjusted for age, gender, smoking status, alcohol consumption, education, and body mass index (BMI); and Model III included adjustments for all covariates mentioned in this study. Additionally, we assessed the joint effect of PA volume and intensity on cognitive impairment risk using Restricted Cubic Splines (RCS). Specifically, participants were stratified into four groups based on the proportion of MVPA and light physical activity (LPA), respectively. The relationship between the PAV in each group and the risk of cognitive impairment was then analyzed using restricted cubic spline modeling. Finally, we analyzed subgroups based on age (< 65 years; ≥ 65 years), gender (male; female), and residence status (rural areas; urban areas) separately. All statistical analyses were conducted using STATA 14 and R software, with a p-value of less than 0.05 considered statistically significant.

## Results

### Demographic characteristics

A total of 10,174 middle-aged and elderly individuals (men: 47.8%; women: 52.2%) were included in the final statistical analysis of this study. As shown in Table 1, significant differences ($P < 0.05$) were observed between insufficient PA and PAV ≥ 600 middle-aged and older adults in terms of age, gender, place of residence, BMI, education, alcohol consumption, hypertension, heart disease, stroke, and incidence of cognitive impairment. However, there were no significant differences ($P > 0.05$) between male and female subjects regarding community environment, duration and frequency of PA, and place of residence. (see Table 1)

### Relationships between the volume and intensity, and the risk of cognitive impairment

Table 2 shows the relationship between the volume and intensity of PA and the risk of cognitive impairment. Subjects with PAV ≥ 600 MET-min/week exhibited a 14% lower risk of cognitive impairment compared to those with insufficient PA (HR: 0.86, 0.77–0.96). Specifically, PAV 600−1199 MET-min/week (HR: 0.71, 0.52–0.94), 1200−1799 MET-min/week (HR: 0.80, 0.67–0.95), 1800−2999 MET-min/week (HR: 0.62, 0.49–0.77), 3000−5999 MET-min/week (HR: 0.86, 0.75–1.00), and 6000−8999 MET-min/week (HR: 0.79, 0.67–0.93) corresponded to reduced risks of cognitive impairment of 29%, 20%, 38%, 14%, and 11%, respectively. In terms of PA intensity, the risk of cognitive impairment was reduced by 25% and 13% for proportions of 0.25–0.5 (HR: 0.75, 0.60–0.92) and 0.5–0.75 (HR: 0.87, 0.77–0.98), respectively, compared to a proportion of 0–0.25 for MVPA to PAV. Additionally, when compared to a proportion of 0–0.25 for LPA to PAV, the risk of cognitive impairment decreased by 15% and 27% for proportions of 0.25–0.5 (HR: 0.85, 0.76–0.95) and 0.5–0.75 (HR: 0.73, 0.59–0.89), respectively.

### RCS plot of volume and intensity combinations of physical activity in relation to the risk of cognitive impairment

The RCS demonstrates a non-linear quantitative relationship between the combination of PAV and intensity and the risk of cognitive impairment (see Fig 2). Regardless of the PAV, the risk of cognitive impairment was highest when the proportion of MVPA to PAV was between 0% and 25%, and it was lowest when this ratio ranged from 25% to 50%. Additionally, the risk of cognitive impairment was highest when the proportion of LPA to PAV was between 0% and 25%, while it was lowest when this ratio fell within the 50% to 75% range.

**Table 1. Baseline characteristics.**

| Characteristics | Total Participants (n = 10174) | Insufficient PA (n = 1614) | PAV ≥ 600 (n = 8560) | P value |
|---|---|---|---|---|
| Age group (years) | | | | 0.00 |
| <65 | 7831(77.00) | 1138(70.50) | 6693(78.20) | |
| ≥65 | 2343(23.00) | 476(29.5) | 1867(21.80) | |
| Gender | | | | 0.00 |
| Men | 4862(47.80) | 689(42.70) | 4173(48.80) | |
| Women | 5312(52.20) | 925(57.30) | 4387(51.30) | |
| Residence | | | | 0.00 |
| Rural | 7781(76.50) | 1174(72.70) | 6607(77.20) | |
| Urban or other | 2393(23.50) | 440(27.30) | 1953(22.80) | |
| BMI | | | | 0.00 |
| <18.5 | 483(4.70) | 96(5.90) | 387(4.70) | |
| 18.5≤and<25 | 3915(38.50) | 540(33.50) | 3375(39.40) | |
| 25≤and<30 | 2127(20.90) | 343(21.30) | 1784(20.80) | |
| ≥30 | 349(35.90) | 635(39.30) | 3014(35.20) | |
| Education | | | | 0.00 |
| ≤High school | 9966(98.00) | 1587(98.30) | 8379(97.90) | |
| ≥College | 208(2.00) | 27(1.70) | 181(2.10) | |
| Smoking status | | | | 0.09 |
| Current | 3029(29.80) | 450(27.90) | 2579(30.10) | |
| Former | 1027(10.10) | 154(9.50) | 873(10.20) | |
| Never | 6118(60.10) | 1010(62.60) | 6118(60.10) | |
| Drinking status | | | | 0.00 |
| Current | 2741(26.90) | 347(21.50) | 2394(28.00) | |
| Former | 677(6,70) | 106(6.60) | 571(6.70) | |
| Never | 6756(66.40) | 1161(71.90) | 5595(65.40) | |
| Chronic conditions | | | | |
| Hypertension | | | | 0.00 |
| No | 7903(77.70) | 1171(72.60) | 6732(78.60) | |
| Yes | 2271(22.30) | 443(27.40) | 1828(21.40) | |
| Diabetes | | | | 0.11 |
| No | 9612(94.60) | 1511(93.60) | 8101(94.60) | |
| Yes | 562(5.50) | 103(6.40) | 459(5.40) | |
| Heart disease | | | | 0.02 |
| No | 8986(88.30) | 1389(86.10) | 7597(88.70) | |
| Yes | 1188(11.70) | 225(13.90) | 963(11.30) | |
| Stroke | | | | 0.00 |
| No | 9976(98.10) | 1564(96.90) | 8412(98.30) | |
| Yes | 198(1.90) | 50(3.10) | 148(1.70) | |
| Cancer: | | | | 0.60 |
| No | 10072(99.00) | 1596(98.90) | 8476(99.00) | |
| Yes | 102(1.00) | 18(1.10) | 84(1.00) | |

PA – physical activity; PAV – Physical activity volume.

**Table 2. Relationships between PA volume and intensity, and the risk of cognitive impairment.**

| Variables | Events/Total N | Model 1 | Model 2 | Model 3 |
|---|---|---|---|---|
| Physical activity volume | | | | |
| Insufficient PA | 395/1614 | ref | ref | ref |
| PAV ≥ 600 MET-min/week | 963/8560 | 0.86(0.77-0.96) | 0.85(0.76-0.95) | 0.86 (0.77-0.96) |
| Insufficient PA | 395/1614 | ref | ref | ref |
| PAV 600–1199 MET  -min/week | 48/294 | 0.68(0.50-0.91) | 0.70(0.51-0.93) | 0.71(0.52-0.94) |
| PAV 1200–1799 MET-min/week | 189/1104 | 0.65 (0.54-0.77) | 0.79(0.66-0.94) | 0.80(0.67-0.95) |
| PAV 1800–2999 MET-min/week | 91/679 | 0.54(0.43-0.68) | 0.61(0.48-0.76) | 0.62(0.49-0.77) |
| PAV 3000–5999 MET-min/week | 346/1870 | 0.79(0.68-0.91) | 0.85(0.74-0.99) | 0.86(0.75-1.00) |
| PAV 6000–8999 MET-min/week | 245/1288 | 0.82(0.70-0.96) | 0.78(0.66-0.91) | 0.79(0.67-0.93) |
| PAV ≥ 9000 MET-min/week | 762/3325 | 1.13(1.00-1.28) | 0.96(0.845-1.08) | 0.97(0.86-1.10) |
| Physical activity intensity | | | | |
| Proportion of MVPA to PAV | | | | |
| 0-0.25 | 764/3559 | ref | ref | ref |
| 0.25-0.5 | 99/612 | 0.79(0.64-0.97) | 0.74(0.59-0.91) | 0.75(0.60-0.92) |
| 0.5-0.75 | 411/2170 | 0.93(0.83-1.05) | 0.86(0.76-0.97) | 0.87(0.77-0.98) |
| 0.75-1 | 802/3833 | 1.17(1.06-1.30) | 0.97(0.87-1.07) | 0.98(0.88-1.08) |
| Proportion of LPA to PAV | | | | |
| 0-0.25 | 1055/4758 | ref | ref | ref |
| 0.25-0.5 | 411/2170 | 0.77(0.69-0.86) | 0.85(0.75-0.95) | 0.85(0.76-0.95) |
| 0.5-0.75 | 99/612 | 0.65(0.53- 0.80) | 0.73(0.59-0.89) | 0.73(0.59-0.89) |
| 0.75-1 | 511/2634 | 0.74(0.66-0.82) | 0.91(0.81-1.01) | 0.91(0.81-1.01) |

PA – physical activity; PAV – Physical activity volume; MET-min/week - metabolic equivalents-minutes/week; MVPA – moderate to vigorous physical activity; LPA – low-intensity physical activity; ref – reference. Modell:adjust for adjusted for age and gender. Model 2: adjusted for age, gender, smoking status, alcohol consumption, education, and body mass index (BMI). Model 3: adjusted for all covariates, including age, gender, smoking status, alcohol consumption, education, BMI chronic disease conditions.

## Subgroup analyses age, gender, place of residence

In our subgroup analyses, after adjusting for covariates, we observed that, for a proportion of MVPA to PAV ratio 0.25–0.5, middle-aged and older adults aged 65 and older (HR: 0.815, 0.667–0.997), men (HR: 0.725, 0.582–0.904), and individuals residing in urban areas (HR: 0.665, 0.445–0.994) were at significantly lower risk of cognitive impairment compared to individuals younger than 65 (HR: 0.913, 0.783–1.065), women (HR: HR: 0.949, 0.821–1.097), and those residing in rural areas (HR: HR: 0.900, 0.792–1.023). The risk of cognitive impairment was significantly lower for individuals younger than 65 (HR: 0.913, 0.783–1.065), women (HR: 0.949, 0.821–1.097), and those living in rural areas (HR: 0.900, 0.792–1.023). A similar trend was observed for the proportion of LPA to PAV 0.5–0.75. Additionally, for the volume of PA, a MET-min/week range of 1800−2999 was associated with a significantly lower risk of cognitive impairment in men (HR: 0.503, 0.319–0.794) compared to women (HR: 0.694, 0.533–0.905) (Fig 3; S1 Table).

## Discussion

This study found that PAV ≥ 600 MET-min/week significantly reduces the risk of cognitive impairment. Additionally, PAV 1800−2999 MET-min/week and the proportion of LPA to PAV 0.5–0.75 (the proportion of MVPA to PAV 0.25–0.5) PA volume and intensity may be more effective in reducing the risk of cognitive impairment in middle-aged and older adults. Furthermore, our findings suggest that individuals aged 65 and older, particularly men, should maybe place a greater emphasis on PA intensity.

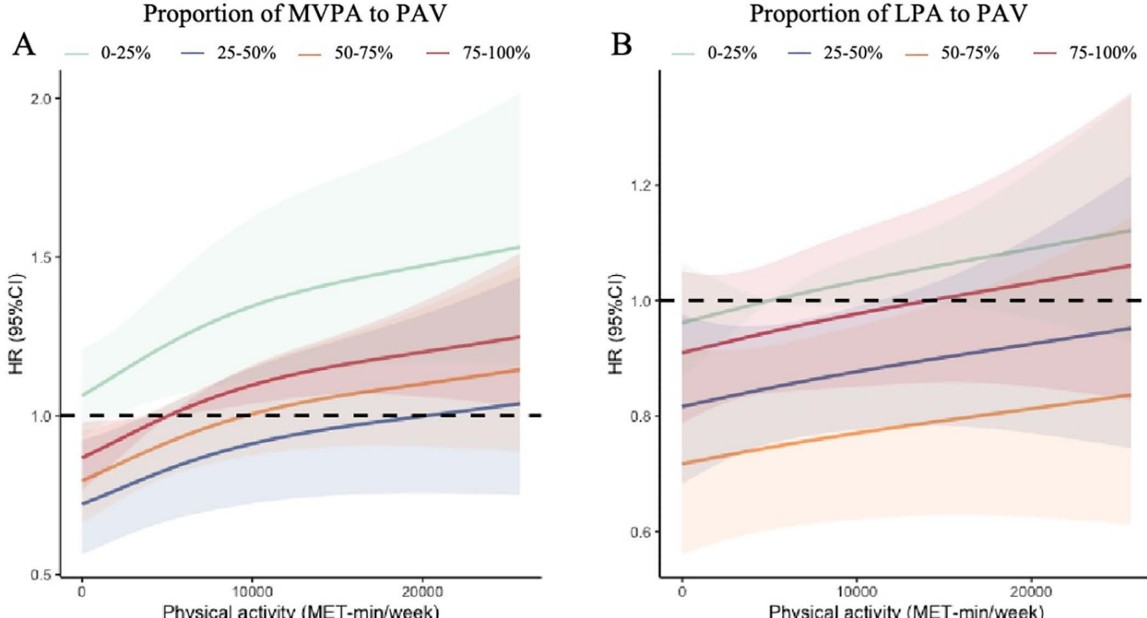

**Fig 2. Relationships between PA volume and intensity combinations and the risk of cognitive impairment.** PA – physical activity; PAV – Physical activity volume; MVPA – moderate to vigorous physical activity; LPA – low-intensity physical activity.

Our study found that PAV ≥ 600 MET-min/week are effective in lowering the risk of cognitive impairment among middle-aged and older adults, consistent with those of prior research [35,36]. Notably, this study identified that a PAV 1800–2999 MET-minutes/week can more effectively reduce the risk of cognitive impairment. Previous studies have largely categorized PAV as either adequate or inadequate [35,37], while our work provides a more nuanced exploration of PAV levels associated with optimal cognitive health outcomes. PA has been shown to enhance brain activity measures, increase prefrontal and temporal grey matter volumes [11], augment cerebral blood volume [38], and improve hippocampal volume [39], thereby promoting cognitive function. Moreover, increased oxygen flow to the brain following PA facilitates neurotransmitter metabolism (including norepinephrine and dopamine), stimulates angiogenesis, enhances synaptic structural integrity [40], and boosts synaptic plasticity. These effects contribute to the prevention of neurodegeneration [41] and enhance the brain's reparative capacity [40]. PA has also been shown to positively affect brain-derived neurotrophic factor (BDNF) [42], a key neuroprotective growth factor that interacts with the tyrosine kinase receptor B to enhance synaptic plasticity and memory [43], thus improving cognitive function [44]. However, excessive PA may compromise the immune system and decrease bone mineral density, among other adverse effects [45], potentially impairing cognitive function in middle-aged and older adults, which may explain that the PA volume ≥9000 MET-min/week group was not significant. Consequently, these can explain the findings of this study that PAV 1800–2999 MET-min/week can be more effective in reducing the risk of cognitive impairment in middle-aged and elderly people.

Regarding the intensity of physical activity. Our findings indicate that PA with a proportion of LPA to PAV 0.5–0.75 (and a proportion of MVPA to PAV 0.25–0.5) is more effective in reducing the risk of cognitive impairment in middle-aged and older adults. MPA has been demonstrated to enhance hippocampal perfusion, hippocampal volume, and cognitive function in healthy older adults [46]. Furthermore, VPA significantly affects acute levels of circulating brain-derived neurotrophic factor and corticospinal excitability, leading to a more substantial reduction in the risk of cognitive impairment [47]. The World Health Organization (WHO) recommends that middle-aged and older adults engage in at least 150 minutes

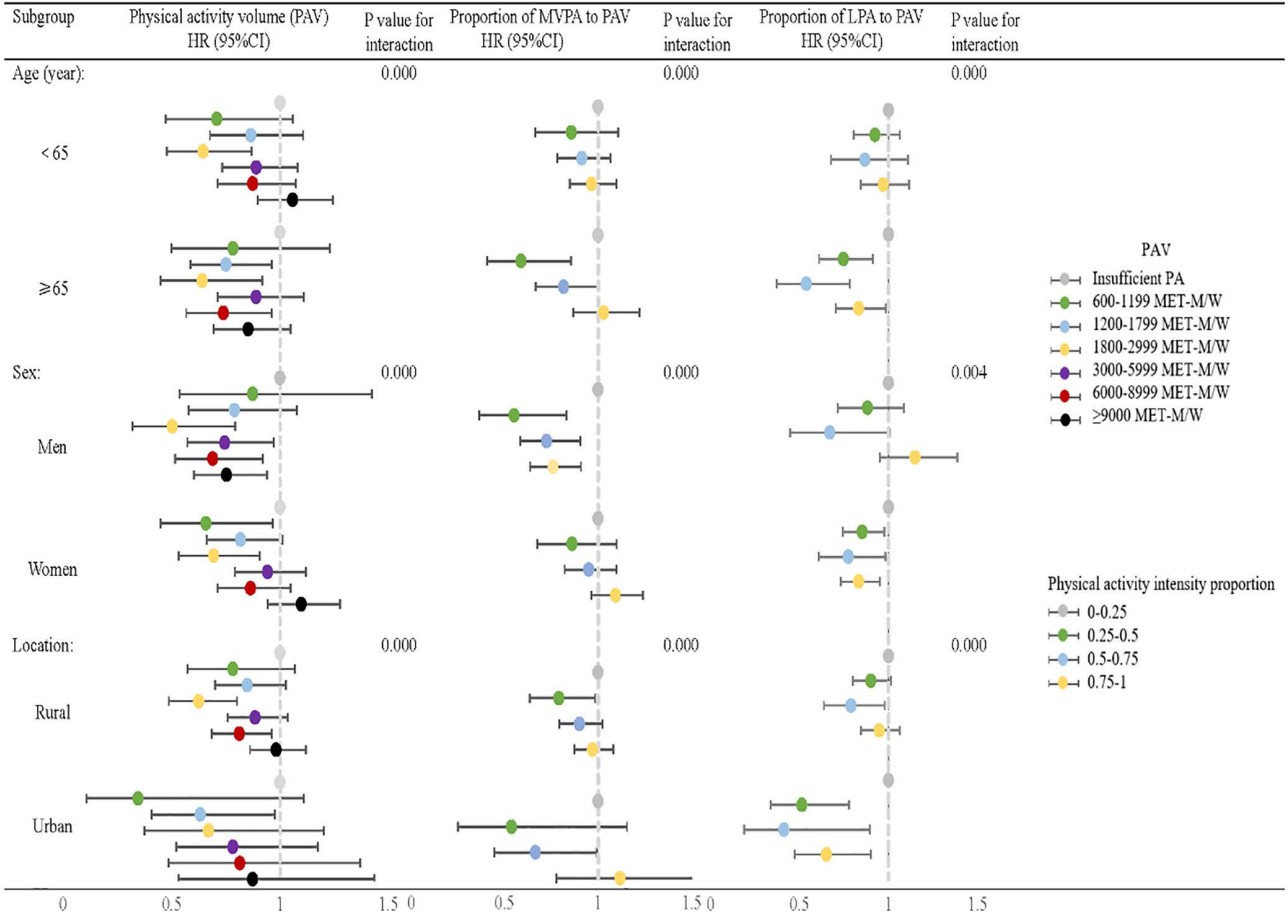

**Fig 3. Sensitivity analyses of the association between PA (volume and intensity) and the risk of cognitive impairment.** PA – physical activity; PAV – Physical activity volume; MVPA – moderate to vigorous physical activity; LPA – low-intensity physical activity.

of MPA, or 75 minutes of VPA, or an equivalent combination of both, each week [13]. However, many middle-aged and older adults fail to meet these recommendations [48]. Additionally, A British study found after a 12-month intervention of moderate-to-high intensity PA training in patients with dementia found reported that MVPA may impair cognitive performance and increase the risk of cognitive impairment in the short term [21]. Another study found a significant association between light walking and hippocampal volume, whereas MVPA showed no such association [49]. Furthermore, there may be an inverted U-shaped dose-response relationship between PA intensity and cognitive function [50]. Unlike MVPA, LPA includes casual walking and housework, activities that provide opportunities for cognitive interaction, listening to music, or enjoying the outdoors, which are associated with higher cognitive performance [51,52]. LPA is more sensitive to subtle changes in cerebral vascular function than MVPA [53]. Therefore, policies and strategies for the prevention and management of cognitive impairment should prioritize PA intensity and emphasize the benefits of LPA. WHO recommendations should be followed for at least 150 minutes of MPA or 75 minutes of VPA per week. In addition, multiple small amounts of PA over a short period of time should be promoted to appropriately increase the total daily LPA. The total amount of MVPA should be reduced appropriately for those middle-aged and elderly people with a larger proportion of MVPA.

Regarding subgroup analyses, our findings indicate that individuals over 65 years of age should emphasize PA intensity. Research has demonstrated that aging is associated with a progressive decline in aerobic fitness, strength, and

muscle mass [54]. High-intensity PA recruits larger motor units, enhances the intensity of chemical processes within the muscle, and involves higher levels of neuromuscular engagement compared to LPA [55,56]. Furthermore, we observed that PA intensity significantly influences the risk of cognitive impairment among middle-aged and older adults residing in urban areas. In contrast, rural residents primarily engage in PA related to agricultural and manual labor, whereas urban Chinese residents benefit from more diverse sports facilities [57]. Urban residents often have a specific inclination towards exercising [58], though they typically engage in activities of lower intensity and lack high-intensity PA. This disparity may stem from structural socioeconomic differences between urban and rural areas. Existing studies reveal an inverse urban pattern: higher MVPA levels often correlate with lower socioeconomic status, whereas rural populations demonstrate less pronounced socioeconomic variation across MVPA groups. Finally, our analysis revealed that both the volume and intensity of PA are more beneficial for cognitive function in middle-aged and older adults. Evidence suggests that men may experience neurobehavioral benefits from unique physical activities through immune activation, such as increased parahippocampal volumes, improved visual memory, and processing speed [59]. Additionally, the correlation between PA and brain volume in women is lower than in men [60], and PA is significantly associated with brain atrophy in men, but not in women [61]. Consequently, policies and strategies should focus on enhancing PA intensity for older adults over 65 years of age, particularly those living in urban areas and men. For older people over 65, it may be possible to develop high intensity interval training in safe, and promote easy-to-use home-based PA guidance programs. Urban residents are more inclined to engage in LPA, so it is possible to encourage more participation in LPA by improving the accessibility and diversity of sports facilities.

## Strengths and limitations

Strengths of this study: First, this study was based on a prospective cohort study design with a nationally representative sample; second, we considered several relevant confounders in our statistical analyses. Limitations of this study: Firstly, the findings may not be applicable to other countries as the participants were Chinese only, So the results may not accurately reflect the relationship between variables across cultures, socioeconomic or population groups; secondly, variables such as PA and cognitive impairment were measured by self-report, which may be subject to recall bias, potentially affecting the validity of the findings. It is recommended that future studies incorporate more objective measurement methodologies, such as the use of accelerometers for assessing physical activity levels; thirdly, the diagnosis of cognitive impairment needs to be based on comprehensive clinical evaluation or evidence; fourthly, the study may have reverse causality, namely that individuals with early cognitive decline may have reduced levels of physical activity as a result of cognitive decline, rather than physical activity affecting cognitive function. Fifth, potential dropout bias requires consideration. Participants discontinuing the study may systematically differ from study completers, potentially biasing the estimated association between physical activity and cognitive impairment.

## Conclusion

This study found that PAV ≥ 600 MET-min/week (This equates to about 150 minutes of moderate-intensity or 75 minutes of vigorous-intensity physical activity per week) is associated with a reduced risk of cognitive impairment. PAV 1800−2999 MET-min/week (This equates to 300–450 minutes of moderate-intensity or 150–225 minutes of vigorous-intensity physical activity per week) and maintaining a proportion of LPA to PAV of 0.5–0.75, or a proportion of MVPA to PAV of 0.25–0.5, are more effective in reducing this risk. Additionally, there should be a particular focus on the PA intensity of older adults aged 65 and above, males, and those residing in urban areas.

## Supporting information

**S1 Table. Association between PA (volume and intensity) and the risk of cognitive impairment.**
(DOCX)

## Acknowledgments

This research has been conducted using the CHARLS. We thank the participants of the CHARLS.

## Author contributions

**Conceptualization:** Wei Yin.

**Data curation:** Wei Yin, Yan Han, Hongmei Sun.

**Formal analysis:** Wei Yin, Yan Han, Hongmei Sun.

**Software:** Wei Yin, Hongmei Sun.

**Supervision:** Yan Han.

**Validation:** Yan Han.

**Writing – original draft:** Wei Yin.

**Writing – review & editing:** Hongmei Sun.

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
