## [Decision Letter · Decision Letter 0]

28 Oct 2025

Dear Dr. Sun,

Thank you for submitting your manuscript to PLOS ONE. After careful consideration, we feel that it has merit but does not fully meet PLOS ONE’s publication criteria as it currently stands. Therefore, we invite you to submit a revised version of the manuscript that addresses the points raised during the review process.

We look forward to receiving your revised manuscript.

Kind regards,

Demitri Constantinou, MD

Academic Editor

PLOS ONE

**Journal Requirements:**

1. When submitting your revision, we need you to address these additional requirements. Please ensure that your manuscript meets PLOS ONE's style requirements, including those for file naming. The PLOS ONE style templates can be found at https://journals.plos.org/plosone/s/file?id=wjVg/PLOSOne_formatting_sample_main_body.pdf and https://journals.plos.org/plosone/s/file?id=ba62/PLOSOne_formatting_sample_title_authors_affiliations.pdf 2. Thank you for uploading your study's underlying data set. Unfortunately, the repository you have noted in your Data Availability statement does not qualify as an acceptable data repository according to PLOS's standards. At this time, please upload the minimal data set necessary to replicate your study's findings to a stable, public repository (such as figshare or Dryad) and provide us with the relevant URLs, DOIs, or accession numbers that may be used to access these data. For a list of recommended repositories and additional information on PLOS standards for data deposition, please see https://journals.plos.org/plosone/s/recommended-repositories. 3. PLOS requires an ORCID iD for the corresponding author in Editorial Manager on papers submitted after December 6th, 2016. Please ensure that you have an ORCID iD and that it is validated in Editorial Manager. To do this, go to ‘Update my Information’ (in the upper left-hand corner of the main menu), and click on the Fetch/Validate link next to the ORCID field. This will take you to the ORCID site and allow you to create a new iD or authenticate a pre-existing iD in Editorial Manager. 4. Your ethics statement should only appear in the Methods section of your manuscript. If your ethics statement is written in any section besides the Methods, please move it to the Methods section and delete it from any other section. Please ensure that your ethics statement is included in your manuscript, as the ethics statement entered into the online submission form will not be published alongside your manuscript. 5. We notice that your supplementary table is included in the manuscript file. Please remove and upload it with the file type 'Supporting Information'. Please ensure that each Supporting Information file has a legend listed in the manuscript after the references list. 6. If the reviewer comments include a recommendation to cite specific previously published works, please review and evaluate these publications to determine whether they are relevant and should be cited. There is no requirement to cite these works unless the editor has indicated otherwise. 

Reviewers' comments:

**Comments to the Author**

1. Is the manuscript technically sound, and do the data support the conclusions?

Reviewer #1: Yes

Reviewer #2: Yes

2. Has the statistical analysis been performed appropriately and rigorously?

Reviewer #1: Yes

Reviewer #2: Yes

3. Have the authors made all data underlying the findings in their manuscript fully available?

Reviewer #1: Yes

Reviewer #2: Yes

4. Is the manuscript presented in an intelligible fashion and written in standard English?

Reviewer #1: Yes

Reviewer #2: No

**Reviewer #1:**  This is an interesting paper on physical activity and cognitive impairment in middle and older adults in China. It is a well-organized paper with a good literature review, statistical rigor, and discussion.

A general comment would be that there are a lot of abbreviations within the abstract. It’d be good if you could summarize those to make it more interesting to read, especially the results part.

In the literature review, lines 95-96 cite an article about machine learning and depression, not sure how that is relevant to the sentence there. Please revise that.

Line 125, there is an abbreviation “ID” without a previous explanation. Please review that.

In lines 172-182, phrases like ‘such as’, ‘including…’ are used, and this might indicate there are more variables listed below, so rephrase it (unless there are more variables not mentioned in that paragraph)

**Reviewer #2:**  In general, the language is good. however there are some areas for suggested improvement. minor edits only:

in the abstract: Suggest "....and their combined risk for cognitive impairment", rather than "their combination"

In the introduction, a suggestion to define cognitive impairment in the context of the study from the outset. Define what exactly this study is examining- cognitive impairment or dementia (which is a major neurocognitive disorder) cognitive impairment should ideally be delineated in terms of mild, moderate or severe, which dementia is an example of. NB in line 87 the study appears to examine neurodegenerative disorders.

line 79, " The World Health Organization (WHO) recommends 150-300 minutes of moderate-intensity physical activity per week, 75-150 minutes of vigorous-intensity physical activity, or some equivalent combination of moderate intensity and vigorous-intensity aerobic physical activity per week. " suggest the authors indicate what exactly for? to maintain a good physique/remain healthy/ avoid dementia, prevent various illnesses?

line 106-108: check grammar. suggest "utilized" or motivate for the study adequately.

line 126-129: suggest full stop after the word "baseline" in "There were 489 data with positive cognitive impairment at

127 baseline. then: "Twenty five with missing data in PA and cognitive impairment data at baseline were excluded, and 1487 with missing data in longitudinal cognitive impairment data were excluded, leaving 11262 data."

line 145, revise sentence to indicate what is "calculated as follows"? the MET score? if so, state this clearly.

line 149, suggest "therefore, we elected to divide the results according into two main categories...."

line 156: suggest "domains" or "functions" not "abilities"

line 165: suggest 'similar to" rather than "as" (in comparing the TICS and the MMSE)

line 176: insert "are" between "(BMI)" and categorized"

**Do you want your identity to be public for this peer review?** For information about this choice, including consent withdrawal, please see our Privacy Policy

Reviewer #1: No

Reviewer #2: No

---

## [Author Response · Author response to Decision Letter 1]

10 Nov 2025

Response by Wei Yin and Hongmei Sun et al.

Ref: PONE-D-25-38651

Title: Independent and joint associations of volume and intensity of physical activity on cognitive impairment among middle-aged and elderly Chinese adults: a national longitudinal study

PLOS ONE

Dear Dr. Sun,

Thank you for submitting your manuscript to PLOS ONE. After careful consideration, we feel that it has merit but does not fully meet PLOS ONE’s publication criteria as it currently stands. Therefore, we invite you to submit a revised version of the manuscript that addresses the points raised during the review process.

We look forward to receiving your revised manuscript.

Kind regards,

Demitri Constantinou, MD

Academic Editor

PLOS ONE

HANDLING EDITOR

Comments

Thank you for submitting your manuscript to PLOS ONE. After careful consideration, we feel that it has merit but does not fully meet PLOS ONE’s publication criteria as it currently stands. Therefore, we invite you to submit a revised version of the manuscript that addresses the points raised during the review process.

Response: Many thanks to the editor for your kind words and the chance for revision submission. We have carefully revised it in accordance with all the reviewers’ comments. Below, we provide point-by-point responses to their feedback. The reviewers’ comments are displayed in blue, our responses in black, and additions to the revised manuscript are highlighted in red.

Reviewer #1: PLOS ONE

Ref: PONE-D-25-38651

Title: “Independent and joint associations of volume and intensity of physical activity on cognitive impairment among middle-aged and elderly Chinese adults: a national longitudinal study”

This is an interesting paper on physical activity and cognitive impairment in middle and older adults in China. It is a well-organized paper with a good literature review, statistical rigor, and discussion.

Response: We sincerely thank the reviewer for the insightful and encouraging comments. We truly appreciate your recognition of the novelty and public health significance of our study. We have carefully addressed each of your suggestions and have revised the manuscript to improve the clarity of writing and ensure overall structural completeness. Detailed point-by-point responses are provided below. We hope the revised manuscript and our responses satisfactorily address your concerns and meet your expectations.

Comment 1: A general comment would be that there are a lot of abbreviations within the abstract. It’d be good if you could summarize those to make it more interesting to read, especially the results part.

Response: We thank the reviewer for this valuable comment. We have summarized abbreviations to make it more interesting to read within the abstract. Then, we have provided the full name for each abbreviation upon its first appearance. (Abstracts section, page 2, lines 36-56).

The revised paragraph now reads as follows:

Abstracts

Background: The objective of this study was to explore the longitudinal relationship between the volume and intensity of physical activity (PA) and their combined risk for cognitive impairment (CI).

Methods: The study included 10,174 participants from the 2011-2018 CHARLS cohort. PA and CI were assessed using self-reported questionnaires. Statistical analyses were performed using the Cox regression model.

Results: After adjusting for all covariates, the risk of CI was 14% lower in subjects with physical activity volume (PAV) ≥600 Metabolic Equivalent of Task (MET)-min/week compared to those with insufficient PA (HR: 0.86). The risk was reduced by 38% for subjects with PAV of 1800-2999 MET-min/week (HR: 0.62). Regarding the intensity of PA, the risk of CI was reduced by 25% for a proportion of 0.25-0.5 of (moderate to vigorous PA) MVPA to PAV (HR: 0.75) compared to a proportion of 0-0.25. Regardless of PAV, the risk of CI was lowest when the proportion of moderate to vigorous PA(MVPA) to PAV was 0.25-0.5, and 0.5-0.75 for the proportion of light-intensity physical activity (LPA) to PAV.

Conclusion: The PAV 1800-2999 MET-min/week and maintaining a proportion of LPA to PAV of 0.5-0.75, or a proportion of MVPA to PAV of 0.25-0.5, are more effective in reducing the risk of CI. Policy implications should prioritize tailored physical activity strategies for individuals over 65, emphasizing low-intensity activities, safe high-intensity training, and the development of accessible urban facilities, in line with WHO guidelines.

Comment 2: In the literature review, lines 95-96 cite an article about machine learning and depression, not sure how that is relevant to the sentence there. Please revise that.

Response: We appreciate this suggestion. We have removed the article about machine learning and depression in the literature review, and we have added the new references. (Introduction section, page 4, lines 95-97).

The revised paragraph now reads as follows:

“Introduction

…Survey To the best of our knowledge, specific guidelines for PA to prevent the onset of cognitive impairment remain absent. Furthermore, the causal relationship between PA and cognitive functioning is still a subject of debate in the existing research. One study observed engaging in high-intensity PA may lead to cognitive decline in middle-aged and older adults [20], whereas another study found that moderate-to-high intensity aerobic physical activity did not slow cognitive impairment in people with mild-to-moderate dementia [21]. …”

The relevant certificate involved is listed as follows:

20. Temesgen, W.A., H.Y. Cheng, and Y.Y. Chong, Cognitive function and its longitudinal predictability by intensity of physical activity in Chinese middle-aged and older adults. J Alzheimers Dis, 2025. 103(3): p. 809-820.

Comment 3: Line 125, there is an abbreviation “ID” without a previous explanation. Please review that.

Response: Thanks for your valuable advice. We have revised the abbreviation “ID” to “identity document”. Thanks. (Methods Materials and methods section, page 5, lines 127).

The revised paragraph now reads as follows:

“Methods Materials and methods

Participants and study design

…

Survey data from 2011-2018 were combined by identity document, resulting in a longitudinal survey of 13,263 data. There were 489 data with positive cognitive impairment at baseline. …”

Comment 4: In lines 172-182, phrases like ‘such as’, ‘including…’ are used, and this might indicate there are more variables listed below, so rephrase it (unless there are more variables not mentioned in that paragraph)

Response: Thanks for your valuable advice. We have deleted those (‘such as’, ‘including…’) in the manuscript. (Methods Materials and methods section, page 7, lines 175-184).

The revised paragraph now reads as follows:

“Methods Materials and methods

Covariates

…

According to previous articles [27, 34]�covariates are primarily categorized into the following groups: 1) Personal variables: age, gender (male, female), education level (high school or below, undergraduate or higher), and residence (rural, urban, or other); 2) Health status-related variables: body mass index (BMI) are categorized as follows: underweight (<18.5), normal weight (18.5 ≤ BMI <25), overweight (25 ≤ BMI <30), and obese (BMI ≥30); 3) Lifestyle variables, smoking status (current smoker, former smoker, never smoker) and alcohol consumption (current drinker, former drinker, never drinker); 4) Chronic disease conditions: hypertension (no, yes), hyperlipidemia (no, yes), diabetes (no, yes), and lung disease (no, yes).”

Reviewer #2: PLOS ONE

Ref: PONE-D-25-38651

Title: “Independent and joint associations of volume and intensity of physical activity on cognitive impairment among middle-aged and elderly Chinese adults: a national longitudinal study”

In general, the language is good. however, there are some areas for suggested improvement.

Response: We thank for your valuable comments and suggestions. We have carefully revised the manuscript to enhance its clarity and facilitate the understanding of readers. Our point-to-point responses are presented in the following.

minor edits only:

Comment 1: in the abstract: Suggest "....and their combined risk for cognitive impairment", rather than "their combination"

Response: We thank the reviewer for this valuable comment. We have revised " their combined " to "their combination". (Abstracts section, page 2, lines 37-38).

The revised paragraph now reads as follows:

“Abstracts

Background: The objective of this study was to explore the longitudinal relationship between the volume and intensity of physical activity (PA) and their combined risk for cognitive impairment (CI).”

Comment 2: In the introduction, a suggestion to define cognitive impairment in the context of the study from the outset. Define what exactly this study is examining- cognitive impairment or dementia (which is a major neurocognitive disorder) cognitive impairment should ideally be delineated in terms of mild, moderate or severe, which dementia is an example of. NB in line 87 the study appears to examine neurodegenerative disorders.

Response: We sincerely thank the reviewer for the valuable suggestion. We have added define cognitive impairment in the context of the study in the introduction. In addition, we have deleted the study appears to examine neurodegenerative disorders in line 87. (Introduction section, page 3, lines 65-67).

The revised paragraph now reads as follows:

“Introduction

Cognitive impairment (CI) refers to a significant decline or dysfunction in memory, learning, attention, language, executive function, and other cognitive domains[1], …”

The relevant certificate involved is listed as follows:

1. Petersen, R.C., et al., Mild cognitive impairment: ten years later. Arch Neurol, 2009. 66(12): p. 1447-55.

Comment 3: line 79, " The World Health Organization (WHO) recommends 150-300 minutes of moderate-intensity physical activity per week, 75-150 minutes of vigorous-intensity physical activity, or some equivalent combination of moderate intensity and vigorous-intensity aerobic physical activity per week. " suggest the authors indicate what exactly for? to maintain a good physique/remain healthy/ avoid dementia, prevent various illnesses?

Response: We thank the reviewer for this insightful suggestion. We have revised "The World Health Organization (WHO) recommends 150-300 minutes of moderate-intensity physical activity per week, 75-150 minutes of vigorous-intensity physical activity, or some equivalent combination of moderate intensity and vigorous-intensity aerobic physical activity per week." to "The World Health Organization (WHO) recommends 150-300 minutes of moderate-intensity physical activity per week, 75-150 minutes of vigorous-intensity physical activity, or some equivalent combination of moderate-intensity and vigorous-intensity aerobic physical activity per week to gain health benefits such as improvements of cognitive health".( Introduction section, page 3, lines 82-85).

Comment 4: line 106-108: check grammar. suggest "utilized" or motivate for the study adequately.

Response: We sincerely thank for the valuable comment. We have revised to "utilized". (Introduction section, page 4, lines 108).

The revised paragraph now reads as follows:

“Introduction

…Based on this, this study utilized a nationally representative sample from the China Health and Retirement Longitudinal Study (CHARLS) to explore the longitudinal relationship between volume and intensity of PA and their combined risk for CI.”

Comment 5: line 126-129: suggest full stop after the word "baseline" in "There were 489 data with positive cognitive impairment at 127 baseline. then: "Twenty five with missing data in PA and cognitive impairment data at baseline were excluded, and 1487 with missing data in longitudinal cognitive impairment data were excluded, leaving 11262 data."

Response: We sincerely thank for the valuable comment. We have revised in the manuscript. (Methods Materials and methods section, page 5, lines 128-129).

The revised paragraph now reads as follows:

“Methods Materials and methods

Participants and study design

…Survey data from 2011-2018 were combined by identity document, resulting in a longitudinal survey of 13,263 data. There were 489 data with positive cognitive impairment at baseline. Twenty five with missing data in PA and cognitive impairment data at baseline were excluded, and 1487 with missing data in longitudinal cognitive impairment data were excluded, leaving 11262 data.”

Comment 7: line 145, revise sentence to indicate what is "calculated as follows"? the MET score? if so, state this clearly.

Response: We sincerely thank for the valuable comment. We have revised "calculated as follows" to " The MET score was calculated as follows " in the manuscript. (Methods Materials and methods section, page 6, lines 147-148).

The revised paragraph now reads as follows:

“Methods Materials and methods

Physical activity

…

The MET score was calculated as follows: PAV = 8.0 × duration of vigorous-intensity physical activity (VPA) per week + 4.0 × duration of Moderate-intensity physical activity (MPA) per week + 3.3 × duration of light-intensity physical activity (LPA) per week [28]. …”

Comment 8: line 149, suggest "therefore, we elected to divide the results according into two main categories...."

Response: We sincerely thank for the valuable comment. We have revised to "therefore, we elected to divide the results according into two main categories...." in the manuscript. (Methods Materials and methods section, page 6, lines 152-153).

The revised paragraph now reads as follows:

“Methods Materials and methods

Physical activity

…

According to IPAQ, Insufficient PA is defined as the PAV less than 600 MET minutes/week [29], therefore, we elected to divide the results according into two main categories (0-599 and ≥600) and seven subcategories in terms of MET minutes/week (0-599, 600-1199, 1200 -1799, 1800-2999, 3000-5999, 6000-8999 and ≥9000). …”

Comment 9: line 156: suggest "domains" or "functions" not "abilities"

Response: We sincerely thank for the valuable comment. We have revised to " functions." in the manuscript. (Methods Materials and methods section, page 6, lines 159).

The revised paragraph now reads as follows:

“Methods Materials and methods

Cognitive impairment

…

Cognitive impairment was assessed based

---

## [Editor Report · Decision Letter 1]

2 Dec 2025

Independent and joint associations of volume and intensity of physical activity on cognitive impairment among middle-aged and elderly Chinese adults: a national longitudinal study

PONE-D-25-38651R1

Dear Dr. Sun,

We’re pleased to inform you that your manuscript has been judged scientifically suitable for publication and will be formally accepted for publication once it meets all outstanding technical requirements.

Kind regards,

Demitri Constantinou, MD

Academic Editor

PLOS ONE
---

## [Editor Report · Acceptance letter]

PONE-D-25-38651R1

PLOS One

Dear Dr. Sun,

I'm pleased to inform you that your manuscript has been deemed suitable for publication in PLOS One. Congratulations! Your manuscript is now being handed over to our production team.

Kind regards,

on behalf of

Professor Demitri Constantinou

Academic Editor

PLOS One